# A dynamic code for economic object valuation in prefrontal cortex neurons

Ken-Ichiro Tsutsui[1,*,†], Fabian Grabenhorst[1,*], Shunsuke Kobayashi[1,†] & Wolfram Schultz[1]

Neuronal reward valuations provide the physiological basis for economic behaviour. Yet, how such valuations are converted to economic decisions remains unclear. Here we show that the dorsolateral prefrontal cortex (DLPFC) implements a flexible value code based on object-specific valuations by single neurons. As monkeys perform a reward-based foraging task, individual DLPFC neurons signal the value of specific choice objects derived from recent experience. These neuronal object values satisfy principles of competitive choice mechanisms, track performance fluctuations and follow predictions of a classical behavioural model (Herrnstein's matching law). Individual neurons dynamically encode both, the updating of object values from recently experienced rewards, and their subsequent conversion to object choices during decision-making. Decoding from unselected populations enables a read-out of motivational and decision variables not emphasized by individual neurons. These findings suggest a dynamic single-neuron and population value code in DLPFC that advances from reward experiences to economic object values and future choices.

[1] Department of Physiology, Development and Neuroscience, University of Cambridge, Downing Street, Cambridge CB2 3DY, UK. * These authors contributed equally to this work. † Present addresses: Division of Systems Neuroscience, Tohoku University Graduate School of Life Sciences, Sendai 980-8577, Japan (K.-I.T.); Department of Neurology, Fukushima Medical University, 1 Hikarigaoka, Fukushima-shi, Fukushima-ken 9601295, Japan (S.K.). Correspondence and requests for materials should be addressed to F.G. (email: fg292@cam.ac.uk).

Rewards are essential goals for economic decisions and behaviour. In natural environments, reward probabilities are often unknown and decision-making requires internal value estimation from recent experience[1–5]. Such value estimates constitute critical elements in reinforcement learning[6] and computational decision theories[7–9]. Although neurophysiological studies uncovered experience-based value signals in different brain structures[1,3,5], key questions about the neural value code remain unresolved.

First, it is unclear how individual neurons encode value estimates as input for decision mechanisms. Biologically realistic decision models use separate value inputs for different choice objects that compete through winner-take-all mechanisms[6–9], rather than explicit relative (comparative) valuations. Although object-specific valuations seem computationally advantageous, relative valuations—which can be derived from object-specific values—are frequently observed in human imaging neural population signals[10–15]. Second, although neuronal values were typically referenced to actions in previous studies[1,3,5], decisions are often made between objects. This distinction is significant, as objects constitute the fundamental choice unit in economic theory. Orbitofrontal cortex (OFC) neurons encode economic object valuations when value is explicitly signalled by external cues[16–18]. However, it is unclear whether object value neurons also encode recent reward experiences, as implied by the concept of value construction[2–4,7], and whether they directly convert values to choices, as predicted by computational models[8,9].

Here we recorded the activity of single neurons in the dorsolateral prefrontal cortex (DLPFC) of monkeys performing an object-based foraging task. The DLPFC is implicated in diverse functions including decision-making[19–28], behavioural control[29–34] and reinforcement learning[35,36]. Previous neurophysiological studies showed that DLPFC neurons encode important economic decision variables including reward probability, reward magnitude, effort[19,26], reward and choice history[20,35,36]. DLPFC is also connected to sensory, motor and reward systems[29,37], including parietal cortex and striatum, where experience-based value signals are found[1,3,5], and anterior cingulate cortex, where lesions impair performance based on reward experience[38].

We hypothesized that individual DLPFC neurons encode the construction of values from experience, their formatting into object-specific decision variables, and their conversion to object choices. We tested whether DLPFC neurons encode values of specific choice objects termed 'object values', in analogy to action values[6] and in line with competitive choice mechanisms[6,39,40]. Although a negative finding would not necessarily contradict the role of DLPFC in decision-making, a positive result would lend credence to the neuronal implementation of competitive decision models, similar to previous single-neuron representations of complex decision variables[41]. We elicited valuations in a foraging task that required internal value construction from reward history and encouraged proportional allocation of choices to rewards received from different objects, following Herrnstein's matching law[42]. The task temporally separated valuation from choice and action, allowing us to identify separate neuronal signals for these distinct computational steps. In addition to single-neuron analysis, we used linear decoding to read out values and value-derived decision variables from DLPFC population activity without pre-selecting neurons for value coding.

We show that individual DLPFC neurons dynamically encode the value of specific choice objects as a decision variable. Individual neurons signal both the construction of object values from recently experienced rewards and their subsequent conversion to object choices. Population decoding demonstrates a dynamic readout of additional value-derived variables not encoded by individual neuron, which meet the motivational and decision requirements of different task stages. This dynamic object value code—characterized by single-neuron convergence of valuation, learning, and decision signals and flexible population readout—may support DLPFC's signature role in adaptive behaviour.

## Results

**Object-based foraging task.** Two monkeys performed in a foraging choice task in which the probability of receiving a reward from each of two options varied dynamically and in an unsignalled manner across trials. In each testing session, two visual objects (A and B) served as choice targets (Fig. 1a). The animal made a saccade to its object of choice and received either a drop of liquid reward or no reward depending on the object's reinforcement schedule. Left-right object positions varied randomly trial-by-trial. During blocks of typically 50-150 trials, each object was associated with a base reward probability according to which a reward was assigned on every trial. Rewards remained available until the animal chose the object. Thus, the instantaneous reward probability for a particular object increased with the number of trials the object was not chosen; it fell back to base probability after each choice of the object. Under such conditions, an effective strategy is to repeatedly choose the object with the higher base probability and only choose the alternative when its instantaneous reward probability has exceeded the base probability of the currently sampled object[2,43]. Global behaviour in such tasks usually conforms to the matching law[42], which states that the ratio of choices to two alternatives matches the ratio of the number of rewards received from each alternative[2–5].

Thus, to maximize reward income, the animals had to track changes in block-wise reward probabilities and local fluctuations owing to the matching task design. This required keeping track of the history of recent rewards and object choices. As reward probabilities within blocks varied predictably in a trial- and choice-dependent manner, the animals could internally evaluate and choose between objects before cue appearance on each trial. This task design, in combination with randomized trial-by-trial cue positions, allowed us to test for neuronal encoding of object values and choices before action selection.

**Matching behaviour and object value model.** Across sessions, both animals conformed to the matching law by allocating their object choices according to the number of rewards available from each object (Fig. 1b). In a representative session, the animal continuously matched its local choice ratio to the current reward ratio (that is, the number of received rewards), and readily adapted to block-wise changes in base probabilities (Fig. 1c). Thus, the animals behaved meaningfully, according to predictions from Herrnstein's matching law, which validated the foraging task as a model for neuronal object valuation and decision processes.

Base reward probabilities and instantaneous probabilities were not externally cued but required learning and continual updating. Thus, internally constructed, subjective value estimates likely guided the animals' choices. To examine neuronal value coding, we estimated these internal values using established approaches[2,4]. Logistic regression determined how the history of past rewards on each object influenced current choices. As matching also required occasional switching between objects, we incorporated a term for choice history[4]. Subjective values for specific choice objects estimated in this manner likely constituted the main decision variable for the animals, which we call 'object value'.

We derived value estimates by convolving object-specific reward and choice histories with filters, obtained from logistic regression, that assigned higher weight to more recent rewards and choices[2–4]. We summed weighted reward and choice histories for each object to obtain scalar, single-trial measures of object value. Filter weights were derived by fitting a logistic regression based on reward and choice history to the animals' choices. The resulting filter shapes (Fig. 1d) resemble those found in previous studies[2–4], with declining absolute weights as a function of past trials.

Choices in a representative session were well described by the value model: model-derived choice probability closely tracked local and block-wise fluctuations in the animal's behaviour and value estimates followed block-wise and local reward income fluctuations (Fig. 1e). For model validation, we performed an out-of-sample prediction with filter weights derived from separate data. This confirmed that object values predicted trial-by-trial choices (Fig. 1f) and that object value difference fitted the animals' choice probabilities (Fig. 1g).

While the value difference between objects likely directed choices, the sum of object values may have been an important motivational influence irrespective of choice direction. Such 'net value' effects are critical in goal-directed behaviour and have previously been shown to influence performance[44]. We tested whether value sum was related to the animals' motivation, measured by trial initiation times (key touch latency). Multiple regression confirmed value sum as main determinant of trial initiation time: the animals' initiated trials faster when value sum was high (Fig. 1h,i), consistent with a motivational effect due to overall reward expectation. By contrast, saccadic reaction times during choice were influenced by the absolute (unsigned) value difference (Supplementary Fig. 1), consistent with previous studies and theoretical models that relate absolute value difference to decision difficulty and confidence[28,45,46].

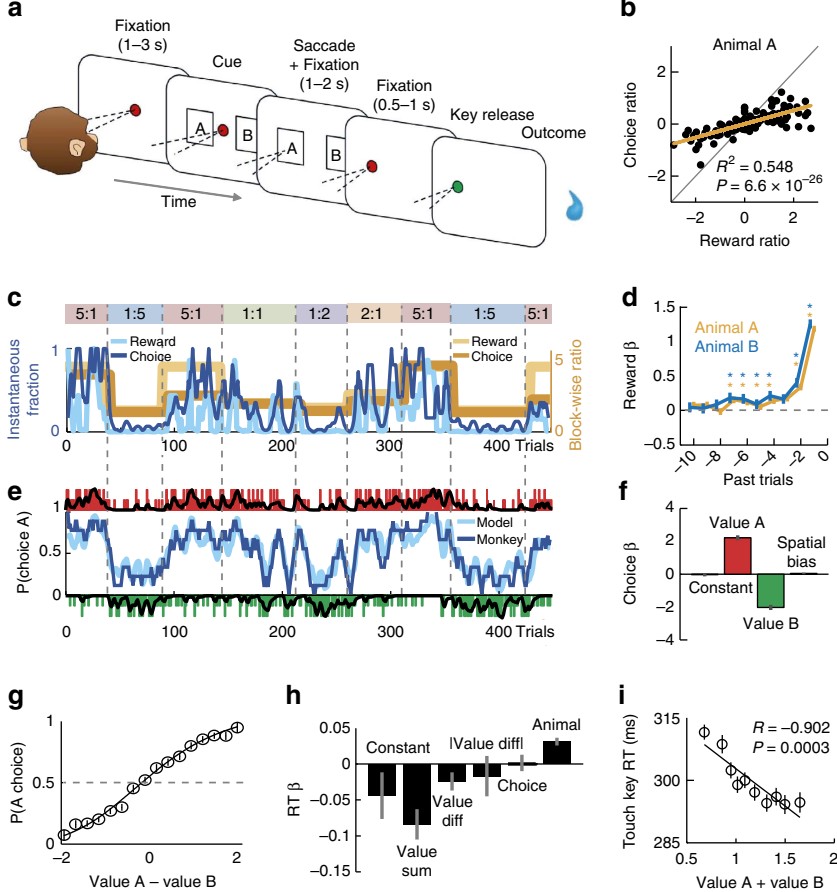

**Figure 1 | Foraging task and matching behaviour. (a)** Object-based foraging task. **(b)** Relationship between log-transformed choice and reward ratios across sessions. The base reward probability ratio was significantly related to the choice ratio (animal A: 11,040 trials from 139 sessions, $t_{137} = 13.09$, linear regression; data were similar for animal B: $R^2 = 0.740$, $P = 4.7 \times 10^{-20}$; 5306 trials from 65 sessions, $t_{63} = 13.37$). **(c)** Behaviour in one example session: choices tracked relative reward probability. Dark blue curve: instantaneous fraction (7-trial running average) of object A choices. Light blue curve: instantaneous fraction of rewards received from object A. Yellow curves: block-wise reward (light) and choice (dark) ratios. Coloured boxes indicate block durations, numbers indicate reward ratios: object A to B. **(d)** Filters used to generate subjective object values: influence of past rewards on current-trial choice. Filters represent logistic regression weights derived from independent behavioural data (animal A: 5,520 trials, d.f. = 5,499; animal B: 2,653 trials, d.f. = 2,632). * $P < 0.05$. **(e)** Model choices closely tracked the animal's choices (same session as in (**c**)). Dark/light blue curve: running average of animal/model choices for object A. Vertical bars: animal choices for objects A (red) and B (green); long bars indicate rewarded trials. Black traces: subjective object values derived from weighted and summed reward and choice histories. **(f)** Out-of-sample prediction of choices from values. Logistic regression weights for object A value ($P = 1.3 \times 10^{-22}$, $t$-test) and object B value ($P = 7.1 \times 10^{-20}$) and cue position (non-significant; coefficients pooled over animals and sessions; 102 sessions, $t$-test, d.f. = 99). **(g)** Decision function relating value difference to choice probability (data pooled over animals and sessions). **(h)** Multiple regression of trial initiation times (key touch latencies) on value sum and covariates (12,358 trials, d.f. = 12,352). Only value sum and animal coefficients were significant (both $P < 0.0001$, $t$-test). **(i)** Single linear regression of trial initiation time on value sum (plot constructed by binning trials according to value sum and then determining reaction time means; data pooled over animals and sessions). Error bars show s.e.m.

Taken together, the animals' choices were well described by object value estimates that were internally constructed and continually updated from reward and choice histories. While value difference was suited to direct choices towards specific objects, value sum reflected the animals' overall motivation.

**Encoding of object value in single DLPFC neurons.** We conceptualize object value analogous to action value[6] as a decision variable that signals the value of specific choice alternatives as suitable input to competitive choice mechanisms. A neuronal response encoding object value should (i) signal value in time to inform the animal's choice, (ii) signal the value of one choice object but not of alternative objects, and (iii) signal value on each trial, irrespective of whether the object is chosen or not. Multiple linear regression analysis determined whether neuronal responses encoded object values according to these criteria while factoring out other task-related variables and testing for alternative (relative) decision variables. Our main conclusions are based on statistical tests within this regression framework; in addition, we plot activity time-courses and single linear regressions to illustrate effects.

The activity of the DLPFC neuron in Fig. 2 fulfilled our criteria for object value coding, as determined by multiple regression analysis. Before appearance of the choice cues, a phasic response leading up to the cue period reflected the current value of object A, with higher activity signalling lower value (Fig. 2a). True to the object value definition, pre-cue activity reflected the value of object A but not of object B (Fig. 2b); no trial period showed a

significant relationship to object B value. Activity was better explained by object value than by object choice (Fig. 2c,d, non-significant choice coefficient). Our experimental design precluded relationships to object position or left-right action in pre-cue periods, as confirmed by non-significant regression coefficients (Fig. 2d). As a further test of object-specificity, we adopted a classification approach based on the angle of regression coefficients in value space (see Methods)[44]. This resulted in a classification scheme of responses into absolute (object) value or relative (sum/difference) value coding depending on the polar angle ($\theta$) of coefficients in value space (coloured areas). This approach confirmed that the neuronal response coded the absolute value of object A (Fig. 2e). Thus, the neuron's pre-cue activity signalled the value of a specific choice object, irrespective of whether the object was chosen.

Among 205 DLPFC neurons with 1222 task-related responses in different task periods ($P < 0.005$, Wilcoxon test), 119 neurons (58%) had value-related activity as indexed by a significant value regression coefficient ($P < 0.05$, multiple regression, Supplementary Table 1). Analysis of different fixed time windows throughout the trial showed that value activity occurred in all task-phases, including pre-cue periods before the animals indicated their choice (Fig. 3a,b). Crucially, visual stimulation and eye position in pre-cue periods were restricted by constant fixation requirement; therefore, these activities did not reflect external sensory information but an internal valuation process. Fixation was also required following the animal's saccade choice until the reward period. In addition, cue position and saccade choice direction were included as covariates in all regression

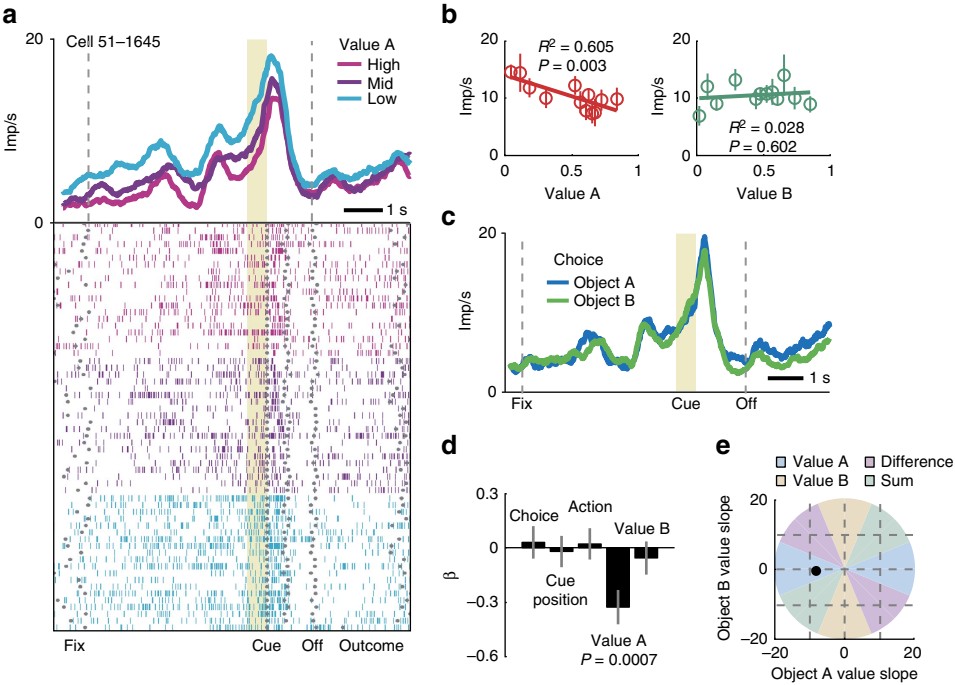

**Figure 2 | A single DLPFC neuron encoding object value.** (**a**) Peri-event time histogram of impulse rates, aligned to cue onset, sorted into terciles of object value (derived from our behavioural model). Raster display: ticks indicate impulses, rows indicate trials; grey dots indicate event markers (labelled below graph; Off: cue offset). Pre-cue activity leading up to the current-trial choice in the cue period reflected the value of object A. Visual stimulation and eye position were constant in this period; thus, the activity pattern reflected an internal valuation process based on reward and choice history and was not due to sensory or motor variables. Yellow shaded period (500 ms before cue onset) was used for analysis. (**b**) Linear regression of pre-cue impulse rate on the value of object A and object B (12 equally populated value bins from 133 trials; d.f. = 11). (**c**) Independence of activity from object choice. Same data as in (**a**) sorted by trial-specific object choice. (**d**) Coefficients obtained from fitting a multiple linear regression model to pre-cue impulse rate. Only the value of object A explained a significant proportion of variance in impulse rate ($t_{127} = -3.48$, $t$-test). All coefficients were estimated simultaneously. (**e**) Plot of regression coefficient in value space using an axis-invariant classification that categorizes neurons as coding object value (value A or value B) or coding relative value (value difference, value sum) based on the polar angle in value space. Error bars show s.e.m.

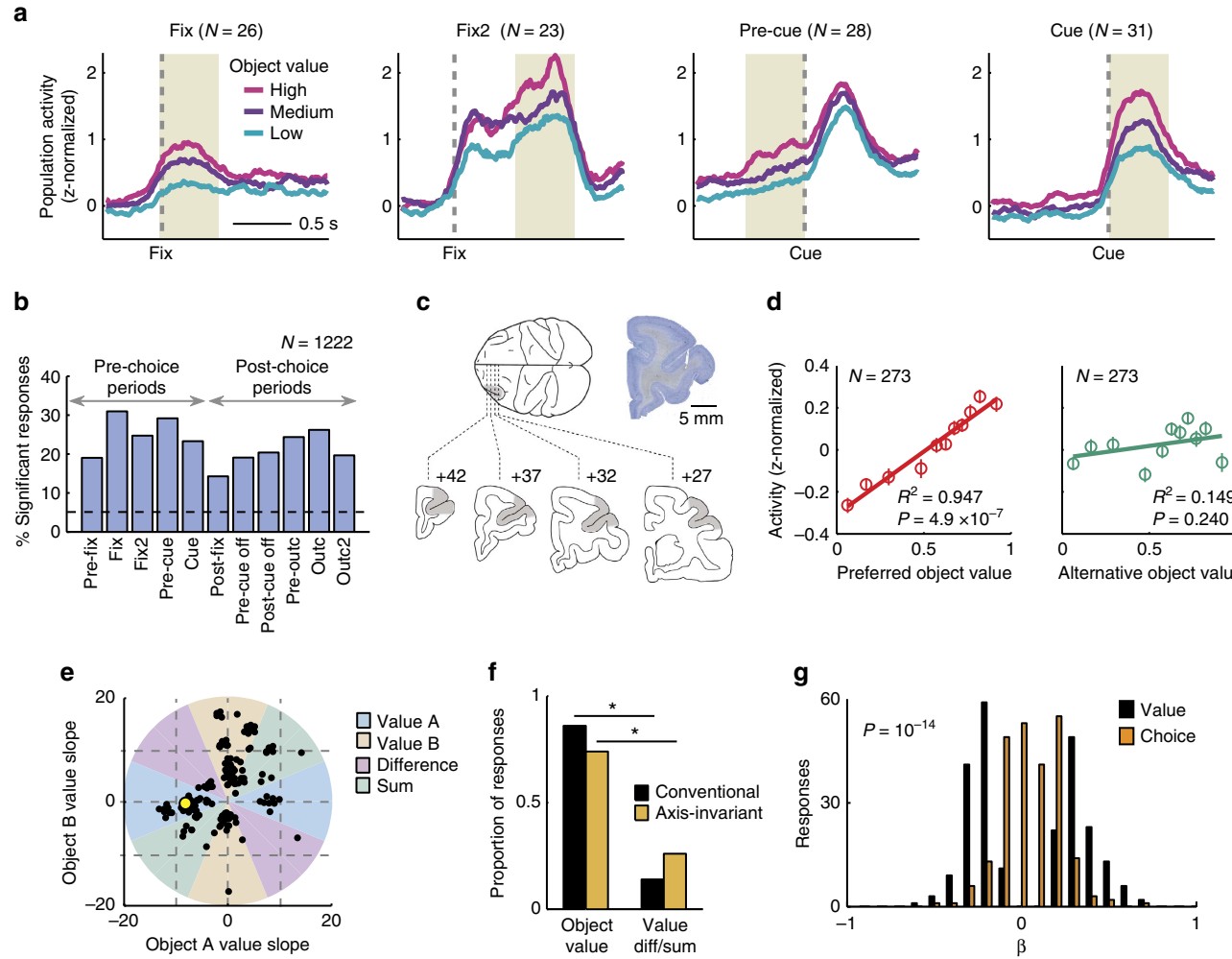

**Figure 3 | Characteristics of value coding in DLPFC neurons.** (**a**) Mean normalized activity for value responses in different trial periods (identified by multiple linear regression, $P < 0.05$, calculated for each response), sorted into value terciles. Yellow shaded periods were used for analysis. N refers to number of responses. (**b**) Percentages of value responses (among task-related responses) in different trial periods. (**c**) Recording locations in upper and lower principal sulcus banks. Numbers indicate anterior-posterior distance from inter-aural line. Sections taken from a stereotaxic atlas[64]. Inset: photomicrograph of a cresyl violet-stained coronal section of the frontal lobe in monkey A. The lesion in the principal sulcus marks a typical electrode track. (**d**) Object-specificity of value coding. Linear regression of population activity (273 value-coding responses taken across task periods) on object value for preferred and alternative object. Data points indicate means of 11 equally populated value bins ± s.e.m. (**e**) Classification of value responses (across task periods) based on position of regression coefficients in value space. The figure shows the classification into object value coding or relative (sum/difference) value coding depending on the polar angle ($\theta$) of coefficients in value space (coloured areas), calculated as four-quadrant arc-tangent of the coefficients. Yellow data point: example neuron from Fig. 2. (**f**) Proportion of value coding responses across task periods reflecting object value (only value A or value B) and responses reflecting relative value (value sum/difference). Black/orange bars: results obtained from conventional multiple regression and axis-invariant method. *$P < 0.05$ ($\chi^2$-test). (**g**) Distribution of regression coefficients (273 value-coding responses; fixed-window analysis across all task periods) for object value and choice. Most value responses had non-significant choice coefficients (Kolmogorov-Smirnov test).

analyses. Sliding window regressions confirmed a substantial number of DLPFC neurons with value-related activity (Supplementary Fig. 2, Supplementary Table 2) and showed that many value signals occurred early in trials around fixation spot onset. Thus, value signals in the DLPFC neuronal population occurred in time to influence object-based decision processes.

Additional tests substantiated the statistical significance of value coding: the observed distribution of value coefficients was significantly different from a distribution based on randomly shuffled data, and shifted towards lower negative and higher positive values (Supplementary Fig. 3). The proportion of significant value coefficients was higher than expected by chance ($P < 0.0001$ binomial test); false positive rate in shuffled data was lower than 5%. Of 273 significant value coefficients (239 individual responses), 131 had a positive sign, implying

higher activity with higher value, and 142 had a negative sign ($P = 0.273$, binomial test, Supplementary Fig. 3). Equal numbers of neurons and responses were found related to object A value and object B value (136/137 responses significant for object A/B, 81 neurons significant for both objects). The neurons were recorded from the upper and lower banks of the principal sulcus, confirmed by histology (Fig. 3c, Supplementary Fig. 4). Thus, a substantial number of DLPFC neurons showed value-related responses. We next show that many of these responses satisfied our criteria for object value coding.

**Object specificity and choice independence of value signals.** An object value response should reflect the value of one specific object without reflecting the value of other objects. True to this

criterion, the majority of value-related activities (205/239, 85.8%) were object-specific without coding value for the other object, as assessed by population activity and significance of value coefficients (Fig. 3d, Supplementary Fig. 3, Supplementary Table 3). Significantly fewer activities coded value for both objects (34/239, 14.2%, $P < 0.001$, $z$-test), which indicated that relative value coding—that is, a relationship to the value sum or difference—occurred in a minority of neurons. An alternative test of object-specificity used a classification approach based on the angle of regression coefficients in value space[44]. Fitting a simpler model that contained only regressors for object A value and object B value (equation (4)) resulted in 168 responses with significant overall model fit ($P < 0.05$, $F$-test; Fig. 3e). Classification into object value and relative value was based on the polar angle ($\theta$) of coefficients in value space. The classification was invariant to the axis choice of value coefficients (see Methods). This axis-invariant method has been suggested to provide a fairer classification into absolute and relative value signals, and can yield different results compared to conventional regressions[44]. However, in our data set of DLPFC neurons, this alternative analysis confirmed our original result: value-related responses were predominantly object-specific; 124 responses were classified as coding object value (74%); 44 responses as coding relative value ($P < 0.001$, $z$-test). Among relative value-coding responses, 35 responses coded value sum (21%) and 9 responses coded value difference (5%). Thus, different analysis approaches confirmed object-specificity of value coding in DLPFC neurons (Fig. 3e,f).

True to the concept of a decision variable, object value signals should occur on trials when the object is chosen and on trials when the object is not chosen. The majority of value-related responses satisfied this criterion by not showing a significant choice coefficient (206/239, 86%, Supplementary Table 3). Distributions of value and choice coefficients in value-coding responses differed significantly, with minor overlap (Fig. 3g). Although both value and choice coding occurred in pre-cue periods, the proportion of pre-cue value responses was significantly higher than that of choice responses ($P = 10^{-7}$, $z$-test). Thus, value coding preceded choice coding in our task.

Our regression model could often not be improved by adding value × choice interaction terms ($P < 0.05$, partial $F$-test): many value-related responses (158 of 1222 task-related responses, 13%) had non-significant value × choice interaction coefficients (compared to 206 choice-independent value responses in our main regression, 17%). Object value responses were also not explained by chosen value coding (Supplementary Fig. 5, Supplementary Table 4).

Randomized cue positions precluded coding of left-right cue position or action before the cue period as confirmed by less than 5% significant coefficients. Following cue onset, a large proportion of DLPFC neurons encoded spatial cue position and left-right action (Supplementary Table 1, Supplementary Fig. 2), reproducing known effects in DLPFC[34,47]. Some of these post-cue responses coded spatial cue position and action jointly with value (Supplementary Table 3). Thus, in addition to pure object value coding, some hybrid responses coded value in conjunction with other task-relevant variables.

Overall, 98 of 611 task-related pre-cue responses (16%) met our strictest criteria for object value coding: value coding for one specific object with insignificant coefficients for the alternative object and insignificant choice coefficient. Taken together, these results show that a substantial proportion of DLPFC neurons coded object value in time to inform the animal's choice and in compliance with formal criteria for a decision variable.

**Action value control.** Optimal behaviour in the foraging task required tracking the value of visual objects rather than of left-right actions. Nevertheless, we also examined whether DLPFC responses reflected action value, as found previously[22,23]. We recalculated our behavioural model by fitting a logistic regression to the animals' left-right choices, based on action and action-reward history[4,5]. Despite providing an inferior fit compared to the object value model, the action value model showed significant filter weights for recent action and action-reward history, typically extending up to two trials into the past. We used the resulting action values as regressors for neuronal activity in supplementary analyses.

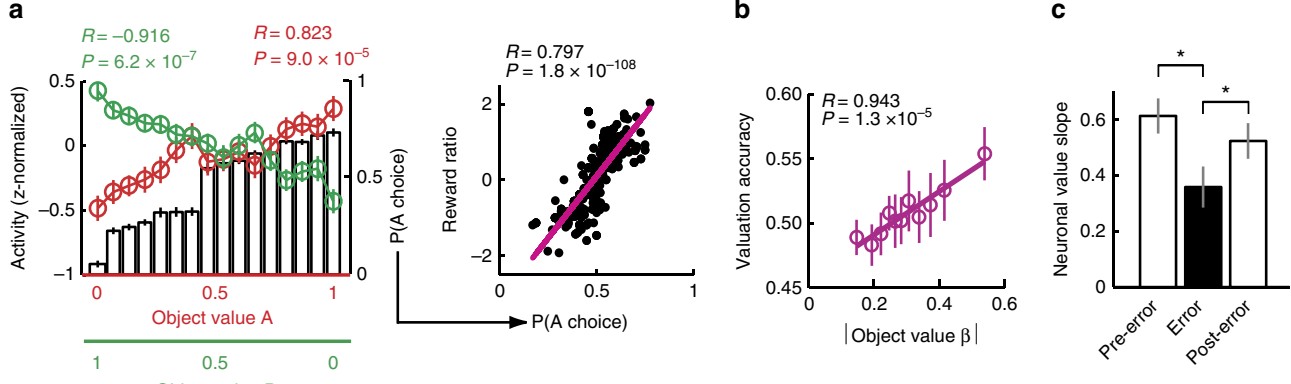

**Figure 4 | Neuronal object values and matching behaviour. (a)** Neuronal object values provide a suitable foundation for local choice probability and global matching behaviour. Left: Average value-related activities for object A (red) and B (green) tracked local choice probability (bars), calculated for discrete 16 value bins (d.f. = 14). Right: Local choice probabilities, binned according to object value and aggregated over trial blocks, were related to experienced reward ratios, consistent with matching behaviour. **(b)** Neuronal sensitivity to object value is related to behavioural valuation accuracy. Unsigned neuronal regression coefficients for value coding plotted against behavioural 'valuation accuracy' (defined as session-specific correlation between estimated values and true reward probabilities). Stronger neuronal value coding predicted more accurate behavioural valuation. Valuation accuracy was also related to better choices (that is, choosing the object with higher true reward probability, $R = 0.178$, $P = 0.0037$, linear regression). **(c)** Relationship between neuronal object values and performance. The strength of neuronal value coding dropped transiently when the animals would commit an error and subsequently increased on the next correct trial. Linear regressions of normalized population activity on object value, calculated across 273 value-coding responses (18,117 trials), separately for pre-error, error and post-error trials. *$P < 0.05$, $t$-test for dependent samples. Data in all plots are taken across all task periods.

Including action values alongside object values in the same model resulted in 165 responses (of 1222 task-related responses, 13%) related to object value but not action value, and 97 responses (8%) related to action value but not object value. The total number of responses related to object value was significantly higher than that for action values (257 versus 192, $P < 0.01$, $z$-test). In a stepwise regression, 171 responses were uniquely explained by object value compared to 126 responses uniquely explained by action value ($P = 0.0053$, $\chi^2$-test). Thus, object value was the more important variable in direct comparisons, even when it competed with action value in the same regression model.

**Behavioural relevance of neuronal object values**. If neuronal object values in DLPFC provided a basis for local choices and global matching behaviour, they should be related to the animals' behaviour. We tested this prediction as follows.

First, to test behavioural relevance at the level of local choice probabilities, we compared the average activity of object value responses for a given value level with the corresponding local behavioural choice probability. As the value of a given object increased, the probability of choosing that object also increased, consistent with our behavioural model (Fig. 4a). Average object value activity for given value levels closely followed local choice probabilities, with opposing trends for value responses related to different objects (left panel). These local choice probabilities were in turn suitable to generate global matching behaviour, as their aggregate over a given session reflected the animals' experienced reward ratio in that session (right panel). Thus, neuronal object values, observed at a local timescale of individual trials, provided a suitable basis for global matching behaviour.

Second, we tested whether the strength of neuronal object value coding was related to the animals' matching performance. We measured the animal's 'valuation accuracy' as the session-specific correlation between object values and the true, trial-by-trial object reward probabilities given by the base probabilities and reinforcement schedule. We then regressed this behavioural valuation accuracy on the neuronal value coding strength (the session-specific slope of the relationship between neuronal activity and value). The strength of neuronal value coding explained variation in valuation accuracy: stronger neuronal value coding was associated with more accurate reward probability estimates (Fig. 4b). In turn, more accurate probability estimates led to a higher proportion of optimal choices, that is, choosing the option with higher momentary reward probability ($R = 0.197$, $P = 0.0011$, linear regression, $N = 205$ sessions from both animals). Thus, stronger neuronal value coding correlated with accurate valuation and better performance.

Finally, if neuronal object values are behaviourally relevant, they should fluctuate with local, trial-by-trial performance, including errors. In a population analysis, we identified trials on which the animal committed an error (for example, failed to release the touch key or broke fixation) and regressed neuronal activity on object value across value-coding neurons. Immediately before error trials, population activity was significantly related to object value (Fig. 4c, 'Pre-error'). The strength of this relationship dropped on subsequent trials when the animals would commit an error ('Error'), and reappeared on the trial following the error ('Post-error'). By contrast, raw impulse rates were not significantly different between error and non-error trials (all $P > 0.1$, Wilcoxon test). Thus, neuronal object value coding transiently declined on error trials, suggesting a relationship with performance fluctuations.

**Single-neuron conversion from experience to object value**. Our behavioural analysis showed that the animals' choices were based on object values that were internally constructed from recent reward history and choice history, which constitute precursors for object values. Consistent with previous findings in DLPFC neurons[20,23,35], direct regression of activity on these history terms showed significant numbers of responses related to last object choice (87/1222, 7%), last action (78/1222, 6%), last outcome (111/1222, 9%) and last object choice × last outcome (78/1222, 6%, Supplementary Fig. 6). The percentage of responses related to the interaction between last action and last outcome (a control variable in our study) did not exceed chance level. Supplementary regression with value and history terms as covariates (Supplementary Table 5) showed that history variables did not account for object value responses (292 significant value coefficients compared to 273 in our main model; 105 value responses (36%) showed non-significant history terms). However, the coding of reward and choice history alongside object value could suggest that individual DLPFC neurons reflect the trial-by-trial construction and updating of object value from recent experience. Such value construction is predicted by our behavioural model, which constructs value from weighted reward and choice history.

A significant number of DLPFC neurons showed dynamic coding transitions consistent with the hypothesized value construction. Across DLPFC neurons, a substantial number were sensitive to both value and last-trial information (113/205, 55%, sliding regression). Early in trials, these neurons encoded past rewards and past choices before encoding a scalar, current-trial value signal (Fig. 5a, Supplementary Fig. 6). We identified 77 neurons (37%) that encoded both last-trial information and value in pre-cue periods. Among them, 47 neurons (61%; 23% of all recorded neurons) encoded last-trial information before encoding current-trial value. The occurrence of such neurons was significantly higher than expected by chance ($P = 1.8 \times 10^{-7}$, binomial test).

If neuronal object values are updated based on last-trial information, individual neurons should have matching selectivity for last-trial information and current-trial value. That is, a neuron encoding current-trial value for one specific object should encode whether that object was chosen on the last trial. We confirmed this prediction by relating the (signed) coefficients for last-trial object choice to those for current-trial value: coefficients for the last-trial choice of object A correlated positively with current-trial value coefficients for object A ($R = 0.938$, $P = 2.7 \times 10^{-23}$, linear correlation, Fig. 5b) and negatively with coefficients for object B ($R = -0.969$, $P = 5.1 \times 10^{-39}$). Such matched neuronal selectivity seems consistent with updating object values from last-trial experience.

These results indicate that DLPFC neurons frequently encoded transitions from last-trial information to current-trial object value. Thus, activity in individual DLPFC neurons appeared to reflect the construction and updating of object values.

**Single-neuron conversion from object value to object choice**. We showed above that a significant number of neurons had responses in specific task epochs that signalled formal object value, without signalling object choice. Across task epochs, however, many neurons exhibited dynamic value-to-choice transitions in the sense that object choice signals followed earlier value signals. The existence of such coding transitions in DLPFC neurons matches the presumed flow of information during decision-making[8,9].

The neuron in Fig. 5c exhibited a value-to-choice conversion: value coding in the fixation period preceded later choice coding in the pre-cue period. This conversion is consistent with a process that transforms an object value input to an object choice output

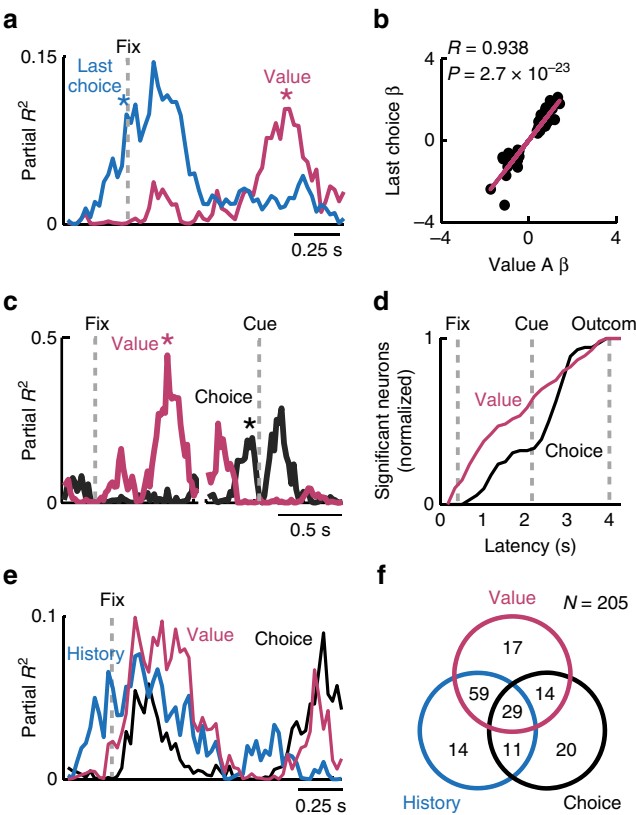

**Figure 5 | Value construction and choice conversion in DLPFC neurons.**
(**a**) A single DLPFC neuron with pre-cue activity reflecting last-trial choice before reflecting current-trial object value. Coefficients of partial determination (partial $R^2$) obtained from multiple regression model applied with a sliding window. Asterisks indicate coding latencies, that is, first time points at which activity was significantly related to a variable. (**b**) Last-trial object choice and current-trial object value were coded with reference to the same object in individual neurons. Neuronal value coefficients for object A ($N = 49$ coefficients, d.f. $= 47$) plotted against coefficients for choice history (defined as last-trial object A choice) for all responses with conjoint value and choice history encoding. A corresponding negative relationship was found for object B value coding ($N = 63$ coefficients, d.f. $= 61$). (**c**) A single DLPFC neuron with pre-cue activity reflecting object value before reflecting object choice. (**d**) Comparison of coding latencies for value and choice. Cumulative record of coding latencies obtained from sliding window regression. Each curve was normalized to the total number of neurons significant for that variable (value: $N = 119$ neurons; choice: $N = 74$ neurons). (**e**) A single DLPFC neuron with pre-cue activity reflecting transitions from last-trial history variables to current-trial object value, and from object value to upcoming object choice. (**f**) Summary of neurons with significant coding of value, history (reward, choice, reward × choice) and choice and their conjunctions, obtained from sliding window regression.

during decision-making. As the neuron's activity did not subsequently reflect cue position or action, it could not by itself instruct action selection but resembled an abstract, action-independent decision process. Other neurons showed dynamic coding transitions that directly converted value to action. We also found neurons exhibiting conversions from object choice to cue position and action (Supplementary Fig. 7), similar to recently reported DLPFC neurons[21]. Critically, although cue position and action signals were related to externally observable events, object value and object choice signals reflected an internal decision process.

Among 95 neurons with pre-choice value coding, the majority (77 neurons, 81%) subsequently coded additional variables.

Specifically, substantial numbers of neurons converted object value to object choice (58/124 value responses, 46%, fixed window-analysis), left–right action (74/124 responses, 60%), or spatial object position (38/124 responses, 31%), with some neurons coding more than one additional variable. By contrast, fewer value neurons (19%, $P < 10^{-13}$, binomial test) either coded no additional variable or coded additional variables prior to value. Given the percentages of significant value and choice coefficients in pre-choice periods, value-to-choice transitions occurred significantly more frequently than chance ($P < 10^{-11}$, binomial test). Across neurons with value or choice coding, value signals appeared significantly earlier compared to choice coding (Fig. 5d, $P < 0.005$, Wilcoxon rank-sum test).

In summary, the critical parameters for decision-making in the foraging task—reward and choice history, object value, and object choice—were dynamically encoded in DLPFC, often converging in single neurons (Fig. 5e,f). A large fraction of DLPFC neurons encoded object value and value precursor variables without encoding choice (90/205, 44%), consistent with the formal object value concept. However, a significant proportion of neurons (29/205, 14%) also combined all three variables. These coding transitions appear consistent with the presumed information flow of value construction, object valuation and decision-making.

**Decoding object value from DLPFC population activity.**
Individual DLPFC neurons likely operate in a population, and their collective value signals could potentially be read out by different downstream neurons for different functions. We used a decoding approach to explore the information about value contained in patterns of population activity that were not pre-selected for task-relatedness or value coding (see Methods). We aggregated trial-specific impulse rates across neurons and used linear support vector machines (SVMs) and nearest-neighbour (NN) classifiers to decode object values and related decision variables. In our main results, we focus on the SVM as it typically performed more accurately. For validation, we found that linear SVMs could decode the basic task variables object choice, cue position and action. For example, action (saccade direction) could be decoded from post-cue activity with near-perfect accuracy (98.90 ± 0.17%, $P < 1.2 \times 10^{-91}$, rank-sum test comparison to randomly shuffled data). Time courses of decoding accuracy closely matched those from single-neuron regressions (Fig. 6a; Supplementary Figs 2,8; $R = 0.96$, $P < 3.4 \times 10^{-20}$, correlation across task periods of decoding accuracy with percentages of significant single-neuron regression coefficients). Notably, the choice for a specific object could be decoded with modest but above-chance accuracy in pre-cue periods (53.27 ± 0.98%, $P < 3.6 \times 10^{-11}$, rank-sum test), whereas cue position and action decoding were non-significant before cue onset, confirming the single-neuron findings. These results provided a useful validation of our population decoding approach.

We tested whether the value of specific objects could be decoded also from population activity (all recorded DLPFC neurons, without pre-selection for task-relatedness or value coding), as suggested by the presence of individual object value-coding neurons. Even without pre-selecting neurons, we decoded object value with good accuracy from the whole population (Fig. 6a; Supplementary Fig. 8). As in single neurons, unselected population activity encoded object value in all task periods, most strongly in the pre-cue period (79.1 ± 0.35%, $P < 6.4 \times 10^{-83}$, rank-sum test). As the pre-cue period was a likely time point of decision-making, we explored how population decoding in this period depended on various parameters.

We quantified value decoding capacity in relation to population size. Decoding performance for object value increased

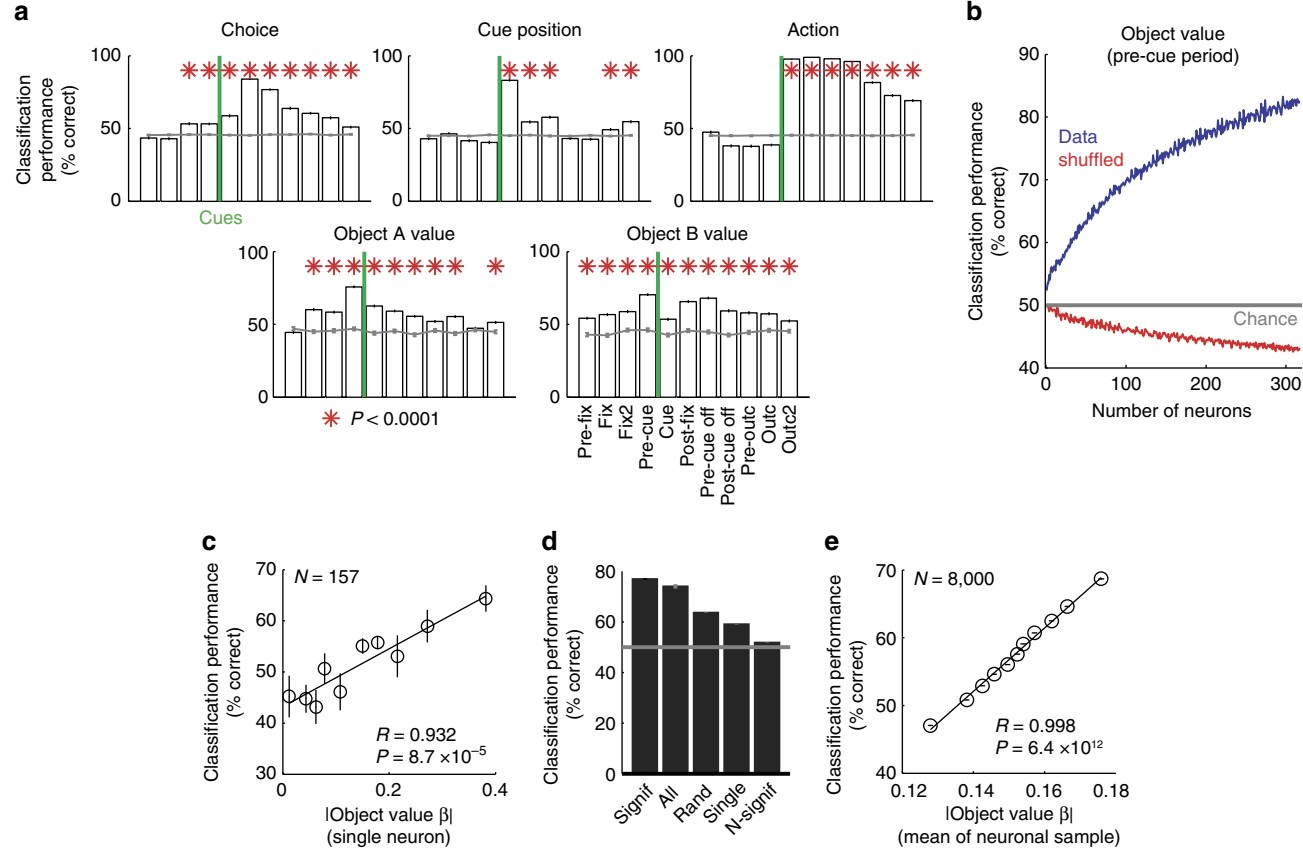

**Figure 6 | Population decoding of object value.** (**a**) Performance of a linear support vector machine classifier in decoding object choice, cue position, action and object value across task periods. Performance was measured as cross-validated classification accuracy (% correct, mean ± s.e.m.) based on single-trial data from all DLPFC neurons that met inclusion criteria for decoding ($N = 166$ for binary variables choice, cue position and action; $N = 157$ for object value terciles). The grey line in each plot indicates mean (± s.e.m) decoding performance from trial-shuffled data. Red asterisks indicate that decoding accuracy significantly exceeded shuffled decoding (rank-sum test). (**b**) Object value decoding performance in the pre-cue period increased with the number of neurons. Data for each neuron number show means (± s.e.m) over 10 random combinations of different neurons. The classifier was trained to decode both object A and B value; thus, data from each neuron ($N = 157$) were sampled twice. (**c**) Object value decoding in individual neurons (in pre-cue period) was related to individual neuron's value sensitivity (object value linear regression slope). (**d**) Object value decoding in different sets of neurons (in pre-cue period), depending on individual neuron's significance of object value regression. Signif: neurons with individually significant object A value regression coefficients (based on randomly chosen subsets of $N = 20$); All: neurons that met inclusion criteria for decoding ($N = 157$); Rand: randomly selected neurons irrespective of object value significance ($N = 20$); Single: single-neuron decoding for all neurons that met inclusion criteria ($N = 157$); N-signif: randomly selected neurons excluding those with significant object value coefficients ($N = 20$). (**e**) Relationship between decoding performance and single-neuron value sensitivities, tested over randomly selected neuron subsets (8000 samples randomly drawn without replacement, $N = 20$ per sample). Decoding depended on average single-neuron sensitivity (mean unsigned value regression coefficient, averaged over all 20 neurons in each sample).

systematically with the number of neurons entered into the decoder (Fig. 6b, Supplementary Fig. 8): while decoding for single neurons was close to chance, accuracy increased approximately linearly over the first 100 neurons as more neurons were added up to a maximum. Such steady increase suggested a distributed representation with different neurons carrying partly independent information about value.

We next analysed how coding in an unselected population depended on the value sensitivity of individual neurons. We found a linear relationship between single-neuron value regression slopes and single-neuron decoding accuracy (Fig. 6c): neurons that maximized value differences (higher value slope) enabled better decoding. Indeed, small subsets of individually significant value neurons provided as good a decoding of object value as the whole population (Fig. 6d) and decoding accuracy was significantly related to single-neuron value sensitivity (Fig. 6e; $P < 1.0 \times 10^{-16}$, partial correlation controlling for number of significant neurons, mean activity range, slope variance).

Thus, neurons with high value sensitivity contributed the most to population decoding, with smaller contributions by non-significant neurons.

These results suggested accurate object value decoding from the DLPFC population. Although decoding generally increased with higher neuron numbers, individually significant value neurons contributed most strongly.

**Population decoding of value-derived decision variables.** In addition to object value, we could decode from the unselected population other value variables not emphasized in single neuron responses, including value sum and signed and unsigned value difference (Fig. 7a). Value sum is an important motivational variable related to performance vigour[44] and predicted trial initiation times (Fig. 1h,i). By contrast, signed value difference is the critical quantity for value comparison in decision models (Fig. 1f,g)[6-9], and unsigned value difference relates to decision

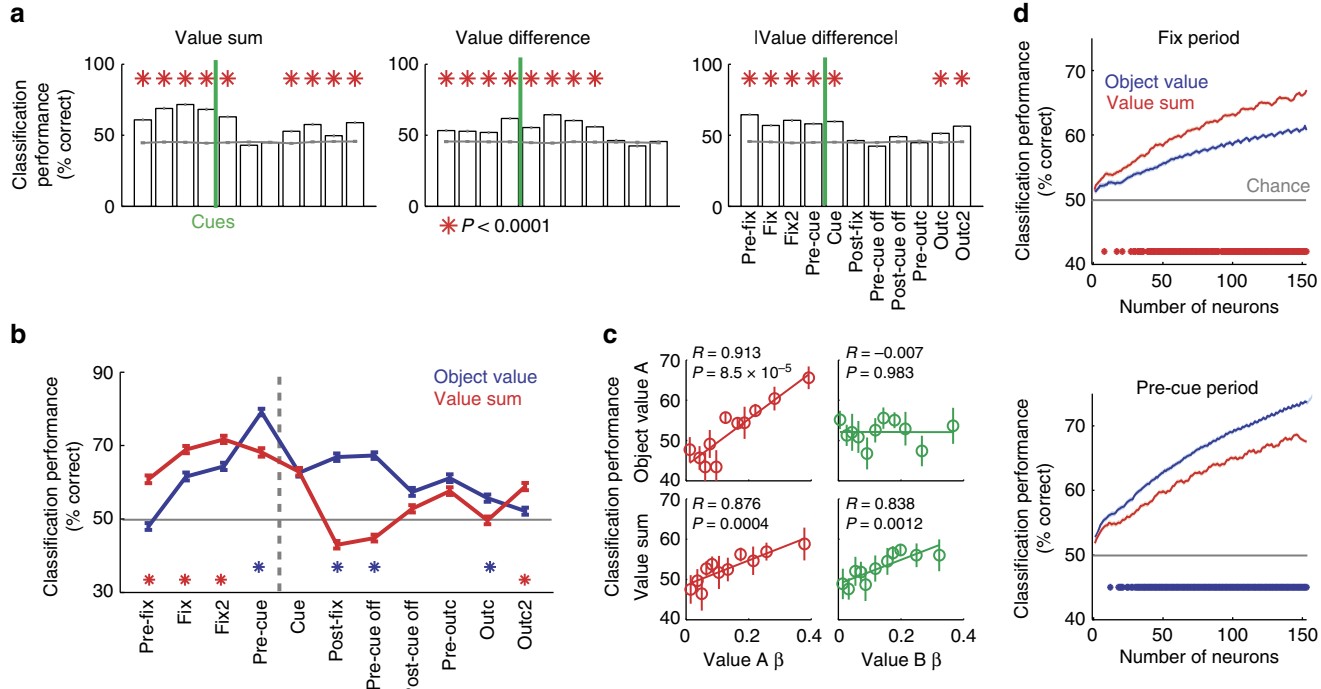

**Figure 7 | Population decoding of decision variables derived from object value.** (**a**) Performance (% correct, mean ± s.e.m.) of a linear support vector machine classifier in decoding value sum, signed value difference and absolute (unsigned) value difference in different task periods ($N = 157$ neurons). (**b**) Comparison of decoding performance for object value (blue data) and value sum (red data). The grey horizontal line indicates chance level. Red asterisks indicate significantly higher accuracy for value sum than object value (rank-sum test); blue asterisks indicate significantly higher accuracy for object value than value sum. (**c**) Relationship between decoding accuracy in individual neurons and neurons' standardized object value regression coefficients, shown separately for object value A (upper panels, pre-cue period) and value sum (lower panels, fixation period). (**d**) Decoding performance as a function of neuron number for object value and value sum in fixation period (top) and pre-cue period (bottom). Asterisks indicate significant differences in decoding accuracy for object value and value sum.

difficulty (Supplementary Fig. 1) and decision confidence[28,45,46]. Average decoding accuracy for value sum and unsigned value difference was most pronounced in early task periods and, compared to object value, was lower and less consistent in later task periods (Fig. 7a). Thus, the unselected neuronal population encoded variables that combined values of different objects, including value sum, signed and unsigned value difference.

Among the different task epochs, the fixation period showed significantly higher decoding accuracy for summed rather than individual object value (Fig. 7b, red). In this early period, value sum decoding reflected single-neuron value sensitivities for both objects A and B (Fig. 7c, lower panels), and more neurons in the decoder increased accuracy significantly more for value sum compared to object value (Fig. 7d, Supplementary Fig. 8c). By contrast, the subsequent pre-cue period showed significantly better population decoding for individual rather than summed object value (Fig. 7b, blue). Here, object value decoding for one specific object reflected single-neuron sensitivities only for that particular object (Fig. 7c, upper panels), which was also evident with the benefit derived from more neurons in the decoder (Fig. 7d). Thus, the key decision variables of object value and value sum were best encoded in particular task periods, which matched the different behavioural functions of value sum (initial motivation, Fig. 1h,i) and object value (subsequent decision-making, Fig. 1f,g).

These findings suggested different levels of value coding in the DLPFC that evolved over trial periods and matched the behavioural requirements in different tasks stages. Single DLPFC neurons encoded object value (Fig. 8a). By contrast, activity in an unselected population encoded additional specific and

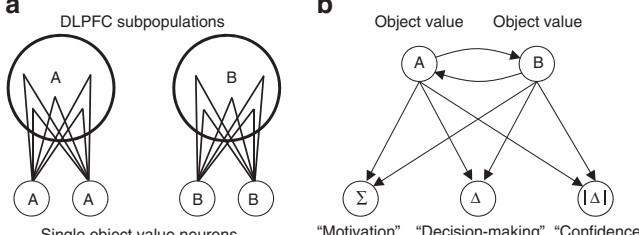

**Figure 8 | Summary of present findings.** (**a**) DLPFC contains subpopulations of neurons that are non-selective for object value but code value precursor variables, including object reward and choice history. We suggest that convergence of these signals onto postsynaptic DLPFC neurons generates the explicit value signals for specific choice objects as observed in single DLPFC neurons in the present study. (**b**) Hypothesized relations between DLPFC object value neurons and population readout, based on the current data. Individual object value neurons learn and update values of specific objects (A, B), signal object-specific values, and convert value to choice signals, as suggested by our single-neuron analyses. Curved arrows indicate a hypothesized competition process via mutual inhibition[7–9] (likely involving pools of inhibitory interneurons[7–9] not shown). Convergence of object value signals onto different downstream neurons could enable readout of value sum ('Motivation'), signed value difference ('Decision-making') and unsigned value difference ('Confidence'), as suggested by our population decoding analyses; these quantities are conceptually linked with specific motivational and decision processes.

well-conceptualized decision variables not represented in single neurons that may make important contributions to distinct behavioural functions (Fig. 8b).

## Discussion

We found that individual DLPFC neurons encoded internal value estimates derived from the fluctuating reward probabilities of specific choice objects. These value signals fulfilled criteria for a decision variable: they were object-specific, distinct from sensory and motor responses, timed to inform decision-making, and independent of current-trial choice. Further, they tracked behavioural performance and followed Herrnstein's matching law and were thus suited to guide the animals' behaviour. Individual DLPFC neurons encoded both the construction of object values from recent reward experience and their subsequent conversion to choice signals. Thus, signals related to these computationally distinct processes converged onto single DLPFC neurons. Object value signals also enabled flexible population readout of decision variables not emphasized by single neurons. Together, our findings suggest that DLPFC realizes a dynamic single-neuron and population value code that reflects the translation of recent reward experiences into economic object values and future choices.

Individual DLPFC neurons mostly encoded value for specific choice objects, rather than relative valuations. We do not argue that explicit object value signals are strictly required for neural decision-making, which likely emerges as a population phenomenon. However, object-specific value coding by single neurons is advantageous computationally because it ensures that value is updated for one specific object but not for others—the key issue of credit assignment in reinforcement learning[6]. It also enables single-neuron conversion from object value to object choice, which ensures unambiguous identification of chosen objects. We found that single DLPFC neurons realized these computational advantages by encoding conversions from experienced rewards to object values and subsequent choices. We suggest that explicit object value signals, rather than relative valuations, would also be observed in situations involving more than two choice objects, although this prediction remains to be tested in future studies.

In addition to explicit single-neuron representations, distributed population codes confer greater flexibility to a neural system for they allow high-accuracy, flexible readout of multiple task variables[48–50]. Consistent with this notion, the population of DLPFC neurons allowed precise decoding of object values (Fig. 8a). The approximately linear increase in decoding accuracy as more neurons were added suggests that neurons carried partly independent value information. Indeed, value sensitivity varied considerably across neurons and population decoding depended on individual neurons' value sensitivities. Such neuronal tuning variation may be advantageous for information processing in associative networks as it can increase storage capacity[50].

Population decoding enabled readout of functionally important variables not emphasized by single neurons. For example, the sum of object values represents a motivational variable suited to calibrate performance vigor[44] and accordingly correlated inversely with the animals' trial initiation times. Consistently, population activity encoded value sum most strongly at trial start. Such a value sum signal arises naturally in biologically realistic decision systems with attractor dynamics, which converge to a choice state faster when value sum is high[7,10]. Flexible population readout of different value variables could be achieved by selective wiring from object-specific value neurons onto different downstream neurons or by dynamically adjustable synaptic connections (Fig. 8b). For example, DLPFC object value subpopulations could provide common inputs to parts of the striatum containing value sum-coding neurons[44]. The additionally observed (although less accurate) population coding of unsigned value difference is predicted by computational decision models[46], and considered a key quantity related to

decision confidence[28,45,46]. (We did not include a behavioural confidence test in our task but unsigned value difference correlated with reaction times). Notably, value difference signals are frequently observed in human neuroimaging population signals, which average across large numbers of neurons[10–15]. Although such techniques successfully localize decision signals across distributed brain systems[10–15], our results suggest that they may not necessarily accurately identify the information encoded by single neurons in a given cortical area.

Two previous studies provided critical evidence that the primate brain computes internal values during matching behaviour[3,5]. Our findings build on this earlier work and offer new insights into the neural basis of value construction. First, value signals during matching in parietal area LIP and striatum are spatially referenced and time-locked to sensory targets or movement onset[3,5]. By contrast, the DLPFC neurons reported here signalled the value of choice objects, rather than actions, irrespective of and prior to action information. Such object-based valuations confer greater flexibility by enabling arbitrary mappings from chosen objects to required actions and by allowing object choices before action information is available. We suggest, following Sugrue and colleagues[3], that abstract, action-independent valuations as uncovered here in DLPFC neurons are computed upstream of LIP and subsequently remapped onto space and action. Our finding that DLPFC neurons convert object values to choices, spatial representations and actions indicate that DLPFC participates in this remapping alongside LIP, although conclusive evidence will require simultaneous recordings from both areas in the same monkeys, performing the same task. Second, in contrast to LIP and striatal neurons, many DLPFC neurons encoded value precursor variables, such as reward and choice history, before encoding value. This could suggest that DLPFC participates actively in the current-trial computation of values from recent reinforcement history. Third, different from striatal action value neurons[5], the presently described DLPFC object value neurons encoded explicit conversions from value to choice. This could suggest a role for DLPFC in the decision process. This interpretation is supported by a recent study[23] showing stronger and earlier action coding in DLPFC compared to striatum, although value-to-choice transitions as shown here were not demonstrated. While the basal ganglia may be important for storing values long term[1,5,51], DLPFC neurons seem important for their construction and conversion to choice on single trials.

The present value-to-choice conversions in single DLPFC neurons are consistent with biologically plausible attractor theories of how decisions arise in neural networks[8,9]. However, the present results cannot determine whether these coding transitions originate in DLPFC or reflect processing in another brain area. This determination will likely require simultaneous recordings from multiple brain systems. Thus, although our experiments cannot directly show that choice computations are performed in DLPFC, our results support the hypothesis[14,24,25] that DLPFC is important for neuronal decision processes.

DLPFC object value neurons resemble offer value neurons in OFC observed during economic choice[16–18] as both types of neuron encode object-specific values irrespective of choice and action. However, whereas separate OFC neurons encode values and choices[17], many DLPFC neurons reported here exhibited dynamic value-to-choice conversions. Further, transitions from reward experience to value as reported here in DLPFC have not been found in OFC. This could suggest that OFC and DLPFC make different contributions to decision-making, or that decision processes differ between choice tasks with explicit value cues and those requiring internal, history-based value construction. The latter interpretation is supported by a recent study with explicitly

cued flavoured juice rewards[21] in which DLPFC neurons showed choice-to-action conversions while apparently only few DLPFC neurons encoded offer values. Our DLPFC object-value signals contrast markedly with explicit relative value (value difference) signals reported in ventromedial prefrontal cortex[52], striatum[44,53,54], and anterior cingulate cortex chosen value signals[55], which could reflect processing differences between DLPFC and these other regions. Although differences in data modelling can contribute to different findings between studies, we confirmed that our results were robust to analysis variations with several regression approaches and population decoding. DLPFC object value neurons also differ from explicit reward prediction by conditioned stimuli[39,51,56–58], as their activity was object-specific, not linked to sensory-motor responses, measured during free choice, and independent of current-trial choice. These features distinguish a genuine decision variable[40] from known reward prediction and reward-modulated sensory-motor activity in DLPFC[59,60]. Finally, although we replicated previously shown DLPFC chosen value signals[21], these were separate from and could not account for object value coding.

In conclusion, our data show that single DLPFC neurons encode reward valuations for specific choice objects based on recent experience. Object value signals complied with criteria for a decision variable, tracked the animals' performance, and followed Herrnstein's classical matching law. Individual DLPFC neurons dynamically encoded conversions from reward and choice history to object value, and from object value to object choice. Thus, DLPFC object value neurons seem well suited to support learning and decision-making in situations requiring internal, experience-based value construction. DLPFC population activity encoded additional value variables not emphasized by single neurons, which could inform motivational and decision processes at different task stages. Together, our data suggest that DLPFC implements a dynamic and computationally flexible object value code, consistent with its signature role in adaptive behaviour.

## Methods

**Animals.** All animal procedures conformed to US National Institutes of Health Guidelines and were approved by the Home Office of the United Kingdom. Two adult male macaque monkeys (*Macaca mulatta*) weighing 5.5–6.5 kg served for the experiments. The number of animals used is typical for primate neurophysiology experiments. The animals had no history of participation in previous experiments. A head holder and recording chamber were fixed to the skull under general anaesthesia and aseptic conditions. Standard electrophysiological techniques permitted extracellular recordings from single neurons in the sulcus principalis area of the frontal cortex via stereotaxically oriented vertical tracks, as confirmed by histological reconstruction. After completion of data collection, recording sites were marked with small electrolytic lesions (15–20 µA, 20–60 s). The animals received an overdose of pentobarbital sodium (90 mg kg$^{-1}$ i.v.) and were perfused with 4% paraformaldehyde in 0.1 M phosphate buffer through the left ventricle of the heart. Recording positions were reconstructed from 50-µm-thick, stereotaxically oriented coronal brain sections stained with cresyl violet.

**Behavioural task.** Each monkey was trained in an oculomotor free-choice task. In every trial, the subject chose one of two objects to which reward was independently and stochastically assigned. Two different abstract pictures served as choice objects (square, 5° visual angle). Each trial started with presentation of a red fixation spot (diameter: 0.6°) in the centre of a computer monitor in front of the animal (viewing distance: 41 cm) (Fig. 1a). The animal fixated the spot and contacted a touch sensitive, immobile resting key at elbow height. An infrared eye tracking system continuously monitored eye positions (ISCAN, Cambridge, MA). During the fixation period at 1.0–2.0 s after eye fixation and key touch, an alert cue covering the fixation spot appeared for 0.7–1.0 s. At 1.4–2.0 s following offset of the alert stimulus, two different visual fractal objects (A, B) appeared simultaneously as ocular choice targets on each side of the fixation spot at 10° lateral to the centre of the monitor. Left and right positions of objects A and B alternated pseudorandomly across trials. The animal made a saccadic eye movement to the target of its choice within a time window of 0.25–0.75 s. A red peripheral fixation spot replaced the target after 1.0–2.0 s of target fixation. This fixation spot turned to green after 0.5–1.0 s, and the monkey released the touch key immediately after colour change.

Rewarded trials ended with a fixed quantity of 0.7 ml juice delivered immediately upon key release. A computer-controlled solenoid valve delivered juice reward from a spout in front of the animal's mouth. Unrewarded trials ended at key release and without further stimuli. The fixation requirements restricted the animals' eye movements in our main periods from trial start to cue appearance and, following the animals' saccade choice, from choice acquisition to reward delivery. This ensured that neuronal activity was minimally influenced by oculomotor activity, especially in our main periods of interest before cue appearance.

According to the basic rule of the matching task, the reward probabilities of objects A and B were independently calculated in every trial, depending on the numbers of consecutive unchosen trials (equation (1)):

$$P = 1 - (1 - P_0)^{n+1} \quad (1)$$

with $P$ as instantaneous reward probability, $P_0$ as experimentally imposed, base probability setting, and $n$ as the number of trials that the object had been consecutively unchosen. This equation implies that reward was probabilistically assigned to the object in every trial, and once a reward was assigned, it remained available until the associated object was chosen. Therefore the likelihood of being rewarded on a target increased as the number of trials performed after the object was last chosen. On the other hand, it stayed at the base probability while the object was repeatedly chosen. The reward probability fell back to the base probability with every choice of that object, irrespective of whether that choice was rewarded or not.

We varied the base reward probability in blocks of typically 50–150 trials without signalling these changes to the animal. The sum of reward probabilities for objects A and B was held constant so that only relative reward probability varied.

**Definition of object value.** We followed an established approach for modelling action value used in previous behavioural and neurophysiological experiments in macaques[2–5]. As the optimal strategy in our task involved tracking the changing values of objects, rather than actions, we formulated the model in terms of object choices rather than action choices. The approach involves fitting a logistic regression model to the animal's trial-by-trial choice data to estimate coefficients for the recent history of received rewards and recently made choices. The resulting coefficients quantify the extent to which the animals based their choices on recently received rewards and made choices for a given option. We used the following logistic regression model to determine the coefficients for reward history and choice history (equation (2)):

$$\log\left(\frac{p_A(i)}{p_B(i)}\right) = \sum_{j=1}^{N} \beta_j^r (R_A(i-j) - R_B(i-j)) + \sum_{j=1}^{N} \beta_j^c (C_A(i-j) - C_B(i-j)) + \beta_0$$

$$(2)$$

with $p_A(i)$ [or $p_B(i)$] as the probability of choosing object A (or B) on the $i$th trial, $R_A$ [or $R_B$] as reward delivery after choice of object A [or B] on the $i$th trial, $C_A$ [or $C_B$] as choice of object A [or B] on the $i$th trial, $N$ denoting the number of past trials included in the model ($N = 10$), $\beta_j^r$ and $\beta_j^c$ as regression coefficients for the effect of past rewards and choices and $\beta_0$ as bias term. The regression model was estimated by fitting regressors to a binary choice indicator function using a binomial distribution with logit link function. The coefficients for reward and choice history from this analysis are plotted in Fig. 1d as reward filters. Within each animal, we used half of the behavioural data set to estimate model coefficients and the remaining half of the data for testing the model. To test the model in an out-of-sample prediction, we used logistic regressions to fit each animal's choices in a given testing session to the corresponding reward and choice histories multiplied with the filter weights obtained from independent data. For this model, we summed the weighted reward and choice histories for each object to obtain measures of object A value and object B value, which constituted our regressors for the out-of-sample prediction. Figure 1f shows the mean coefficients for these object values averaged over both animals and all remaining sessions (random effects analysis). The same object value measures were used as regressors for neuronal data.

**Neuronal data analysis.** We counted neuronal impulses in each neuron on correct trials relative to different task events with 500 ms time windows that were fixed across neurons: before fixation spot (Pre-fix, starting 500 ms before fixation onset), early fixation (Fix, following fixation onset), late fixation (Fix2, starting 500 ms after fixation spot onset), pre-cue (Pre-cue, starting 500 ms before cue onset), cue (Cue, following cue onset), post-fixation (Post-fix, following fixation offset), before cue offset (Pre-cue off, starting 500 ms before cue offset), after cue offset (Post-cue off, following cue offset), pre-outcome (Pre-outc, starting 500 ms before reinforcer delivery), outcome (Outc, starting at outcome delivery), late outcome (Outc2, starting 500 ms after outcome onset).

We first identified task-related responses in individual neurons and then used multiple regression analysis to test for different forms of value-related activity while controlling for the most important behaviourally relevant covariates. We identified task-related responses by comparing activity to a control period (Pre-fix) using the Wilcoxon test ($P < 0.005$, Bonferroni-corrected for multiple comparisons). A neuron was included as task-related if its activity in at least one task period was significantly different to that in the control period. Because the Pre-fixation period served as control period we did not select for task-relatedness in this period and included all neurons with observed impulses in the analysis. We chose the

pre-fixation period as control period because it was the earliest period at the start of a trial in which no sensory stimuli were presented. The additional use of a sliding-window regression approach for which no comparison with a control period was performed (see below) confirmed the results of the fixed window analysis that involved testing for task-relationship. The fixed-window analysis identified the following numbers of task-related responses in the different task periods: Pre-fix: 205, Fix: 84, Fix2: 93, Pre-cue: 96, Cue: 133, Post-fix: 119, Pre-cue off: 110, Post-cue off: 103, Pre-outc: 115, Outc: 103, Outc2: 61.

We next used multiple regression analysis to assess relationships between neuronal activity and task variables. The use of multiple regression was considered appropriate for the present data after testing assumptions of randomness of residuals, constancy of variance and normality of error terms. Statistical significance of regression coefficients was determined using $t$-test with $P < 0.05$ as criterion, and was supported by the results of a bootstrap technique as described in the Results. Our analysis followed established approaches previously used to test for value coding in different brain structures[1,5]. All tests performed were two-sided. Each neuronal response was tested with the following main multiple regression model (equation (3)):

$$y = \beta_0 + \beta_1 \text{ObjectChoice} + \beta_2 \text{CuePosition} + \beta_3 \text{Action} + \beta_4 \text{ObjectValueA}$$
$$+ \beta_5 \text{ObjectValueB} + \varepsilon$$

(3)

with $y$ as trial-by-trial neuronal impulse rate, ObjectChoice as current-trial object choice (0 for A, 1 for B), CuePosition as current-trial spatial cue position (0 for object A on the left, 1 for object A on the right), Action as current-trial action (0 for left, 1 for right), ObjectValueA as current-trial value of object A, ObjectValueB as current-trial value of object B, $\beta_1$ to $\beta_5$ as corresponding regression coefficients, $\beta_0$ as constant, $\varepsilon$ as residual. Object value regressors were defined as described in the previous section. Coefficients for all regressors within a model were estimated simultaneously. Thus, a significant regressor indicated that a significant portion of the variation in neuronal impulse rate could be uniquely attributed to this variable. We followed standard procedures for assessing multicollinearity in multiple regression analysis. We confirmed that variance inflation factors were generally low (Mean $= 1.53 \pm 0.17$ s.e.m.; 99% of VIFs $< 3$; VIFs calculated separately within each neuronal testing session for regression model in equation (3)), indicating that multicollinearity did not affect our statistical analysis.

For the regression analysis shown in Fig. 3e, we fit the following model to the neuronal data (equation (4)):

$$y = \beta_0 + \beta_1 \text{ObjectValueA} + \beta_2 \text{ObjectValueB} + \varepsilon \quad (4)$$

A neuronal response was categorized as value-related if it showed a significant overall model fit ($P < 0.05$, $F$-test). We then projected each value-related response onto the value space given by the regression coefficients for object value A and object value B (Fig. 3e). Following a previous study[44], we divided the value space into eight equally spaced segments of 45° which provided a categorization of neuronal responses based on their polar angle of coefficients in value space. Responses were classified as coding object value ('absolute value') if their coefficients fell in the segments pointing toward 0° or 180° (object value A) or toward 90° or 270° (object value B). Responses were categorized as coding value difference if their coefficients fell in the segments pointing towards 135° or 315° and as coding value sum if their coefficients fell in the segments pointing towards 45° or 225°. This method of classification has been called 'axis-invariant' as its results do not depend on the choice of axis for the regression model, that is, whether the regression model includes separate independent variables for object values A and B or separate independent variables for the sum and differences between object values[44].

We also used a sliding window multiple regression analysis (using the regression model in equation (3)) with a 200-ms window that we moved in steps of 25 ms across each trial. To determine whether neuronal activity was significantly related to a given variable we used a bootstrap approach based on shuffled data as follows. For each neuron, we performed the sliding window regression 1000 times on trial-shuffled data and determined a false positive rate by counting the number of consecutive windows in which a regression was significant with $P < 0.05$. We found that less than 5% of neurons with trial-shuffled data showed more than six consecutive significant analysis windows. In other words, we used the shuffled data to obtain the percentage of neurons with at least one case of six consecutively significant windows. Therefore, we counted a sliding window analysis as significant if a neuron showed a significant ($P < 0.05$) effect for more than six consecutive windows.

**Normalization of population activity.** We subtracted from the measured impulse rate in a given task period the mean impulse rate of the control period and divided by the standard deviation of the control period ($z$-score normalization). Next, we distinguished neurons that showed a positive relationship to object value and those with a negative relationship, based on the sign of the regression coefficient, and sign-corrected responses with a negative relationship.

**Normalization of regression coefficients.** Standardized regression coefficients were defined as $xi(si/sy)$, $xi$ being the raw slope coefficient for regressor $i$, and $si$ and

$sy$ the standard deviations of independent variable $i$ and the dependent variable, respectively. These coefficients were used for Figs 2d, 3g, 4b, 5b, 6c,e, 7c, Supplementary Fig. 1 and Supplementary Fig. 3a,c.

**Population decoding.** We used SVM and NN classifiers to quantify the information contained in DLPFC population activity in defined task periods, following decoding analysis approaches from previous neurophysiological studies[61–63]. The SVM classifier was trained on a set of training data to find a linear hyperplane that provides the best separation between two patterns of neuronal population activity defined by a grouping variable (for example, high versus low object value). Decoding was typically not improved by non-linear (for example, quadratic) kernels. The NN classifier was similarly trained on a set of test data and decoding was performed by assigning each trial to the group of its nearest neighbour in a space defined by the distribution of impulse rates for the different levels of the grouping variables using the Euclidean distance[62]. Both SVM and NN classification are biologically plausible in that a downstream neuron could perform similar classification by comparing the input on a given trial with a stored vector of synaptic weights. Both classifiers performed qualitatively similar, although SVM decoding was typically more accurate. We therefore focus our main results on SVM decoding.

We aggregated $z$-normalized trial-by-trial impulse rates of independently recorded DLPFC neurons from specific task periods into pseudo-populations. We used all recorded neurons that met inclusion criteria for a minimum trial number, without pre-selecting for value coding, except where explicitly stated. For each decoding analysis, we created two $n$ by $m$ matrices with $n$ columns defined by the number of neurons and $m$ rows by the number of trials. We defined two matrices, one for each group for which decoding was performed (for example, high versus low object value, left versus right action and so on). Thus, each cell in a matrix contained the impulse rate from a single neuron on a single trial measured for a given group. Because neurons were not simultaneously recorded, we randomly matched up trials from different neurons for the same group and then repeated the decoding analysis with different random trial matching (within-group trial matching) 150 times for the SVM and 500 times for the NN. We found these numbers to produce very stable classification results. (We note that this approach likely provides a lower bound for decoding performance as it ignores potential contributions from cross-correlations between neurons; investigation of cross-correlations would require data from simultaneously recorded neurons.) We used a leave-one-out cross-validation procedure whereby a classifier was trained to learn the mapping from impulse rates to groups on all trials except one; the remaining trial was then used for testing the classifier and the procedure repeated until all trials had been tested. An alternative approach of using 80% trials as training data and testing on the remaining 20% produced highly similar results[61]. We only included neurons in the decoding analyses that had a minimum number of 10 trials per group for which decoding was performed, and we confirmed that results were very similar when increasing this minimum number to 20 trials.

The SVM decoding was implemented in Matlab (Version R2013b, Mathworks, Natick, MA) using the 'svmtrain' and 'svmclassify' functions with a linear kernel and the default sequential minimal optimization method for finding the separating hyperplane. Decoding could typically not be improved by using radial basis function or quadratic kernels. The NN decoding was performed in Matlab using custom-written code. We quantified decoding accuracy as the percentage of correctly classified trials, averaged over all decoding analyses for different random within-group trial matchings. To investigate how decoding accuracy depends on population size, we randomly selected a given number of neurons at each step and then determined the percentage correct. For each step (that is, each possible population size) this procedure was repeated 10 times. We also performed decoding for randomly shuffled data (shuffled group assignment without replacement) with 1500–5000 iterations to test whether decoding on real data differed significantly from chance. Statistical significance ($P < 0.0001$) was determined by comparing vectors of percentage correct decoding accuracy between real data and randomly shuffled data using the rank sum test[62]. For all analyses, decoding was performed on neuronal responses taken from the same task period. We trained classifiers to distinguish high from low value terciles (decoding based on median split produced very similar results). Notably, even these discretized values fit significantly to choices ($P = 2.4 \times 10^{-6}$, logistic regression), suggesting they were behaviourally relevant.

**Data availability.** The data that support the findings of this study are available from the corresponding author upon reasonable request.

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

## Acknowledgements

We thank Istvan Hernádi for suggestions on the experiment, Armin Lak for discussions, Mercedes Arroyo for histology, and the Wellcome Trust and the Behavioural and Clinical Neuroscience Institute (BCNI) Cambridge for financial support.

## Author contributions

K.T. and W.S. designed the experiments, K.T, F.G. and W.S. developed the theoretical framework, K.T. and S.K. performed the experiments, K.S. performed initial data analysis, F.G. designed and performed the final data analyses, F.G. and W.S. wrote the paper.

## Additional information

**Competing financial interests:** The authors declare no competing financial interests.

DOI: 10.1038/ncomms16175

# Author Correction: A dynamic code for economic object valuation in prefrontal cortex neurons

Ken-Ichiro Tsutsui, Fabian Grabenhorst, Shunsuke Kobayashi & Wolfram Schultz

*Nature Communications* 7:12554 doi: 10.1038/ncomms12554 (2016); Published 13 Sep 2016; Updated 23 Nov 2017.

The previously published version of this Article contained an error in Figure 1. In panel b the $x$ and $y$ axis labels of the scatter graph were inadvertently inverted. This error has been corrected in both the PDF and HTML versions of the Article.

