## [Peer Review File · Nature Communications]

Editorial Note: this manuscript has been previously reviewed at another journal that is not operating a transparent peer review scheme. This document only contains reviewer comments and rebuttal letters for versions considered at *Nature Communications*.

REVIEWERS' COMMENTS:

Reviewer #1 (Remarks to the Author):

This is a revised article, documenting representation of object value by dorsolateral prefrontal neurons, in monkeys trained to perform a foraging task. Prior studies have shown modulation of neuronal activity in the dorsolateral prefrontal cortex (and areas connected with it, such as the orbitofrontal and posterior parietal cortex) by factors related to reward, including value and choice. Here, the monkeys were given the choice between two discrete objects, allowing the authors to examine the representation of object value. The authors have provided convincing answers to the comments of my original review. Additional analyses are now presented, which strengthen the manuscript. Overall, I find the manuscript suitable for publication in *Nature Communications*. I have only a couple of remaining comments:

1. The authors used only two objects, the same across all experiments, to assess object value. Would relative value be expected to increase if more objects were used? The authors should at least consider and discuss this possibility, in the discussion.
2. It is difficult to follow the relationship between neurons and "responses" tested. The authors report 205 neurons with task-related responses (page 7), then move one to report 239 individual responses with significant value coefficients (page 8), then 1222 task-related responses (page 9), then 611 pre-cue responses (page 9). Explain, briefly (in the results or methods), how many "responses" were drawn from each neuron, and how these break down between task conditions.

Point-by-point responses to referees

We have addressed the remaining reviewers' points as described below. The reviewers' comments are shown in italics, our responses are preceded by "Authors' response" and our changes to the paper are shown in red in the manuscript.

Reviewer 1.

This is a revised article, documenting representation of object value by dorsolateral prefrontal neurons, in monkeys trained to perform a foraging task. Prior studies have shown modulation of neuronal activity in the dorsolateral prefrontal cortex (and areas connected with it, such as the orbitofrontal and posterior parietal cortex) by factors related to reward, including value and choice. Here, the monkeys were given the choice between two discrete objects, allowing the authors to examine the representation of object value. The authors have provided convincing answers to the comments of my original review. Additional analyses are now presented, which strengthen the manuscript. Overall, I find the manuscript suitable for publication in Nature Communications. I have only a couple of remaining comments:

1. The authors used only two objects, the same across all experiments, to assess object value. Would relative value be expected to increase if more objects were used? The authors should at least consider and discuss this possibility, in the discussion.

Authors' response: We have addressed this point by including the following new text in the discussion, page 16, paragraph 2:

"We suggest that explicit object value signals, rather than relative valuations, would also be observed in situations involving more than two choice objects, although this prediction remains to be tested in future studies."

2. It is difficult to follow the relationship between neurons and "responses" tested. The authors report 205 neurons with task-related responses (page 7), then move one to report 239 individual responses with significant value coefficients (page 8), then 1222 task-related responses (page 9), then 611 pre-cue responses (page 9). Explain, briefly (in the results or methods), how many "responses" were drawn from each neuron, and how these break down between task conditions.

Authors' response: We have addressed this point by including the following new text in the Results, page 7, paragraph 2, and in the Methods, page 23, paragraph 2:

"Among 205 DLPFC neurons with 1222 task-related responses in different task periods ($P < 0.005$, Wilcoxon test) 119 neurons (58%) had value-related activity as indexed by a significant value regression coefficient ($P < 0.05$, multiple regression, **Supplementary Table 1**)."

"The fixed-window analysis identified the following numbers of task-related responses in the different task periods: Pre-fix: 205, Fix: 84, Fix2: 93, Pre-cue: 96, Cue: 133, Post-fix: 119, Pre-cue off: 110, Post-cue off: 103, Pre-outc: 115, Outc: 103, Outc2: 61."

Additional reviewer comment provided by editor in the manuscript:

The authors have done a good job in highlighting the differences in their results with those of a previous study in area LIP (Sugrue et al., Science, 2004). Frankly, for this issue to be resolved, recordings in the two areas will be necessary in the same monkeys, performing the same task. The authors would be well advised to make this point.

Authors' response: We have addressed this point by including the following new text in the discussion, page 17, paragraph 2:

“Our finding that DLPFC neurons convert object values to choices, spatial representations and actions indicate that DLPFC participates in this remapping alongside LIP, although conclusive evidence will require simultaneous recordings from both areas in the same monkeys, performing the same task.”